# Machine Learning Consensus Clustering of Morbidly Obese Kidney Transplant Recipients in the United States

**DOI:** 10.3390/jcm11123288

**Published:** 2022-06-08

**Authors:** Charat Thongprayoon, Shennen A. Mao, Caroline C. Jadlowiec, Michael A. Mao, Napat Leeaphorn, Wisit Kaewput, Pradeep Vaitla, Pattharawin Pattharanitima, Supawit Tangpanithandee, Pajaree Krisanapan, Fawad Qureshi, Pitchaphon Nissaisorakarn, Matthew Cooper, Wisit Cheungpasitporn

**Affiliations:** 1Division of Nephrology and Hypertension, Department of Medicine, Mayo Clinic, Rochester, MN 55902, USA; charat.thongprayoon@gmail.com (C.T.); supawit_d@hotmail.com (S.T.); pajaree_fai@hotmail.com (P.K.); qureshi.fawad@mayo.edu (F.Q.); 2Division of Transplant Surgery, Mayo Clinic, Jacksonville, FL 32224, USA; mao.shennen@mayo.edu; 3Division of Transplant Surgery, Mayo Clinic, Phoenix, AZ 85054, USA; jadlowiec.caroline@mayo.edu; 4Division of Nephrology and Hypertension, Department of Medicine, Mayo Clinic, Jacksonville, FL 32224, USA; mao.michael@mayo.edu; 5Renal Transplant Program, University of Missouri-Kansas City School of Medicine/Saint Luke’s Health System, Kansas City, MO 64108, USA; napat.leeaphorn@gmail.com; 6Department of Military and Community Medicine, Phramongkutklao College of Medicine, Bangkok 10400, Thailand; 7Division of Nephrology, University of Mississippi Medical Center, Jackson, MS 39216, USA; pvaitla@umc.edu; 8Department of Internal Medicine, Faculty of Medicine, Thammasat University, Pathum Thani 12120, Thailand; 9Department of Medicine, Division of Nephrology, Massachusetts General Hospital, Harvard Medical School, Boston, MA 02114, USA; pitch.nissa@gmail.com; 10Medstar Georgetown Transplant Institute, Georgetown University School of Medicine, Washington, DC 20007, USA; matthew.cooper@gunet.georgetown.edu

**Keywords:** obesity, body mass index, kidney transplant, transplantation

## Abstract

Background: This study aimed to better characterize morbidly obese kidney transplant recipients, their clinical characteristics, and outcomes by using an unsupervised machine learning approach. Methods: Consensus cluster analysis was applied to OPTN/UNOS data from 2010 to 2019 based on recipient, donor, and transplant characteristics in kidney transplant recipients with a pre-transplant BMI ≥ 40 kg/m^2^. Key cluster characteristics were identified using the standardized mean difference. Post-transplant outcomes, including death-censored graft failure, patient death, and acute allograft rejection, were compared among the clusters. Results: Consensus clustering analysis identified 3204 kidney transplant recipients with a BMI ≥ 40 kg/m^2^. In this cohort, five clinically distinct clusters were identified. Cluster 1 recipients were predominantly white and non-sensitized, had a short dialysis time or were preemptive, and were more likely to receive living donor kidney transplants. Cluster 2 recipients were older and diabetic. They were likely to have been on dialysis >3 years and receive a standard KDPI deceased donor kidney. Cluster 3 recipients were young, black, and had kidney disease secondary to hypertension or glomerular disease. Cluster 3 recipients had >3 years of dialysis and received non-ECD, young, deceased donor kidney transplants with a KDPI < 85%. Cluster 4 recipients were diabetic with variable dialysis duration who either received non-ECD standard KDPI kidneys or living donor kidney transplants. Cluster 5 recipients were young retransplants that were sensitized. One-year patient survival in clusters 1, 2, 3, 4, and 5 was 98.0%, 94.4%, 98.5%, 98.7%, and 97%, and one-year death-censored graft survival was 98.1%, 93.0%, 96.1%, 98.8%, and 93.0%, respectively. Cluster 2 had the worst one-year patient survival. Clusters 2 and 5 had the worst one-year death-censored graft survival. Conclusions: With the application of unsupervised machine learning, variable post-transplant outcomes are observed among morbidly obese kidney transplant recipients. Recipients with earlier access to transplant and living donation show superior outcomes. Unexpectedly, reduced graft survival in cluster 3 recipients perhaps underscores socioeconomic access to post-transplant support and minorities being disadvantaged in access to preemptive and living donor transplants. Despite obesity-related concerns, one-year patient and graft survival were favorable in all clusters, and obesity itself should be reconsidered as a hard barrier to kidney transplantation.

## 1. Introduction

Obesity is associated with poor outcomes after kidney transplantation. Obesity-related perioperative transplant concerns include wound-related morbidity, delayed graft function (DGF) acute rejection, increased length of hospital stay, healthcare cost, cardiovascular disease, and diabetes after transplantation [1,2,3,4,5,6]. As a result of these issues, body mass index (BMI) is commonly incorporated in the assessment of kidney transplant candidacy [7,8]. However, when compared to remaining dialysis-dependent, kidney transplantation is significantly associated with better survival and long-term mortality across all body mass index (BMI) groups. Consequently, while BMI is routinely included as a determinant of transplant candidacy, there is no consensus on BMI restrictions for kidney transplantation, and acceptable BMI limits for kidney transplant candidates vary among transplant centers [9].

Existing practice guidelines recommend a supervised weight loss regimen including a low-calorie diet, behavioral therapy, and a physical activity plan to achieve a more optimal BMI prior to kidney transplantation [10]. While most clinicians and patients would accept the potentially increased post-transplant complication risks associated with BMI 30–35 kg/m^2^ patients, those with an even higher BMI, especially BMI > 40 kg/m^2^, are frequently declined for kidney transplantation. Although less than 5% of kidney transplant surgeries are performed for patients with a BMI of ≥40 kg/m^2^, morbidly obese kidney transplant patients are heterogenous, and the effects of BMI on kidney transplant outcomes vary as a result of patient characteristics [8,11,12].

Artificial intelligence has been increasingly applied in the medical field for different sectors. Its application in the form of machine learning (ML) has allowed it to develop tools that improve clinical decision making and individualize patient care [13,14,15,16,17,18]. Unsupervised consensus clustering is a type of ML that can elucidate novel data patterns and distinct subtypes from large data [19,20,21]. By detecting unique similarities and differences in diverse data variables that may have previously gone unnoticed, it can categorize these findings into clinically meaningful clusters [19,20]. The literature supports the use of ML unsupervised consensus clustering analysis. Studies have demonstrated the ability to identify distinct groups that forecast different clinical outcomes [22,23]. Given that data on characteristics of morbidly obese kidney transplant recipients are limited, the application of a ML consensus clustering approach may provide a novel understanding of unique phenotypes of morbidly obese kidney transplant recipients with distinct outcomes to identify strategies to improve their outcomes.

In this study, we analyzed the United Network for Organ Sharing database (UNOS)/OPTN database from 2010 through 2019 using an unsupervised ML clustering approach to identify distinct clusters of morbidly obese kidney transplant recipients and assess clinical outcomes among these unique clusters.

## 2. Materials and Methods

### 2.1. Data Source and Study Population

This study was conducted utilizing the Organ Procurement and Transplantation Network (OPTN)/United Network for Organ Sharing database (UNOS) database. All adult morbidly obese end-stage kidney disease patients receiving kidney-only transplant between 2010 and 2019 in the United States were included. Morbid obesity was defined as having a body mass index (BMI) ≥ 40 kg/m^2^. Patients who underwent combined kidney transplant with other organs were excluded. This study was approved by The Mayo Clinic Institutional Review Board (IRB number 21-007698).

### 2.2. Data Collection

The following recipient-, donor-, and transplant-related variables were abstracted from the OPTN/UNOS database: recipient age, sex, race, BMI, number of kidney transplants, dialysis duration, cause of end-stage kidney disease, comorbidities, panel reactive antibody (PRA), hepatitis C, hepatitis B, and human immunodeficiency virus (HIV) serostatus, Karnofsky performance status index, working income, insurance, US residency status, education, serum albumin, kidney donor type, ABO incompatibility, donor age, sex, race, history of hypertension in donor, kidney donor profile index (KDPI), HLA mismatch, cold ischemia time (CIT), kidney on pump, delay graft function (DGF), allocation type, Epstein-Barr virus (EBV), cytomegalovirus (CMV) status, and induction and maintenance immunosuppression. BMI was calculated as body weight in kilograms divided by the square of height in meters. Missing data of all variables were less than 10% (Appendix A). Missing data were imputed through the multivariable imputation by chained equation (MICE) method [24].

### 2.3. Clustering Analysis

Unsupervised ML was applied by conducting consensus clustering to categorize clinical phenotypes of morbidly obese kidney transplant recipients [25]. A pre-specified subsampling parameter of 80% with 100 iterations and the number of potential clusters (k) ranging from 2 to 10 were used to avoid producing an excessive number of clusters that would not be clinically useful. The optimal number of clusters was determined by examining the consensus matrix (CM) heat map, cumulative distribution function (CDF), cluster-consensus plots with the within-cluster consensus scores, and the proportion of ambiguously clustered pairs (PAC). The within-cluster consensus score, ranging between 0 and 1, was defined as the average consensus value for all pairs of individuals belonging to the same cluster [26]. A value closer to one indicates better cluster stability. PAC, ranging between 0 and 1, was calculated as the proportion of all sample pairs with consensus values falling within the predetermined boundaries [27]. A value closer to zero indicates better cluster stability [27]. The PAC was calculated using two criteria: (1) the strict criteria consisting of a predetermined boundary of (0, 1), where a pair of individuals who had a consensus value > 0 or <1 was considered ambiguously clustered, and (2) the relaxed criteria consisting of a predetermined boundary of (0.1, 0.9), where a pair of individuals who had a consensus value > 0.1 or <0.9 was considered ambiguously clustered [27]. The detailed consensus cluster algorithms used in this study for reproducibility are provided in the Appendix A.

### 2.4. Outcomes

Post-transplant outcomes consisted of patient death, death-censored graft failure within 1 and 5 years after kidney transplant, and acute allograft rejection within 1 year after kidney transplant. We defined death-censored graft failure as the need for dialysis or kidney retransplant, while censoring patients for death or at the last follow-up date reported to the OPTN/UNOS database.

### 2.5. Statistical Analysis

After each morbidly obese kidney transplant recipient was assigned to a cluster using the consensus clustering approach, the comparison of clinical characteristics and post-transplant outcomes among the assigned clusters was performed. The clinical characteristics among the assigned clusters were compared using the Chi-squared test for categorical variables and analysis of variance (ANOVA) for continuous variables. The key characteristics of each cluster were identified using the standardized mean difference between each cluster and the overall cohort with the pre-specified cut-off of >0.3. The cumulative risks of death-censored graft failure and patient death after kidney transplant were estimated using the Kaplan–Meier method and were compared among the assigned cluster using Cox proportional hazard analysis. As OPTN/UNOS only reports whether allograft rejection occurred within one year after kidney transplant but did not specify the occurrence date, the risk of one-year acute allograft rejection was compared among the assigned clusters using logistic regression analysis. The association of the assigned cluster and post-transplant outcomes was not adjusted in multivariable analysis for differences in clinical characteristics because the unsupervised consensus clustering approach purposefully generated clinically distinct clusters. R, version 4.0.3 (RStudio, Inc., Boston, MA, USA; http://www.rstudio.com/, accessed on 21 July 2021), was used for statistical analyses, ConsensusClusterPlus package (version 1.46.0) for consensus clustering analysis, and the MICE command in R for multivariable imputation by chained equation [24].

## 3. Results

A total of 158,367 kidney transplants occurred in the United States from 2010 to 2019. In this group, consensus clustering analysis identified 3204 (2%) kidney transplant recipients with a BMI > 40 kg/m^2^ (Table 1). For this entire cohort, the median BMI was 41.7 (IQR 40.6–43.4) kg/m^2^. Over half of the recipients (54%) had >3 years of time on dialysis; however, 14% were preemptive. Diabetes (31%), hypertension (24%), and glomerular disease (23%) were the most common causes of end-stage kidney disease, and 42% of recipients were diabetic. Sixty-five percent of recipients received standard KDPI (KDPI < 85%) deceased donor kidneys and 31% received living donor kidneys. Only 4% received high-KPDI kidneys. The median CIT was 11.7 (IQR 2.7–18.8) hours, and 28% of recipients experienced DGF. Thymoglobulin was the most commonly used induction medication (54%).

In this cohort of 3204 obese kidney transplant recipients, 5 clinically distinct clusters were identified. Figure 1A demonstrates the CDF plot consensus distributions for each cluster of morbidly obese kidney transplant recipients, where the delta area plot shows the relative change in the area under the CDF curve (Figure 1B). The largest changes in area occurred between k = 4 and k = 6, at which point the relative increase in area became noticeably smaller. As shown in the CM heat map (Figure 1C, Appendix A), the ML algorithm identified consensus matrix k = 2, k = 3, and k = 5 with clear boundaries, indicating good cluster stability over repeated iterations. The mean cluster consensus score was comparable in k = 2, k = 3, and k = 5 (*p* > 0.05) (Figure 2A). Favorable low PACs by both strict and relaxed criteria were demonstrated for k = 5 (Figure 2B). Thus, using baseline variables at the time of transplant, the consensus clustering analysis identified five clusters that best represented the data pattern of morbidly obese kidney transplant recipients.

### 3.1. Clinical Characteristics of Morbid Obesity Clusters

Characteristics of the five clinically distinct clusters are shown in Table 1. According to standardized mean difference, shown in Figure 3A–E, cluster 1 was characterized by recipients who were preemptive (26%) or had a shorter time on dialysis (25% < 1 year dialysis, 29% 1–3 years dialysis). Common causes of kidney disease in cluster 1 recipients included diabetes (33%) and glomerular disease (27%). Cluster 1 recipients were not sensitized (PRA 0%) and were the most likely of the five clusters to receive a living donor kidney transplant (97%).

Cluster 2 recipients were older (median age 58 (IQR 51–64) years) and were most likely to have been on dialysis for more than 3 years. Diabetes was the leading cause of kidney disease (57%) in cluster 2, and 70% of cluster 2 recipients were diabetic. The majority of cluster 2 recipients received deceased donor kidney transplants from non-ECD standard KDPI donors (68%). Median donor age was 49 (IQR 39–55) years, and 42% had a history of hypertension, with 11% of cluster 2 recipients receiving high-KDPI kidney transplants. Median CIT for cluster 2 was 17 (IQR 12–23) hours, 57% of kidneys were placed on machine perfusion, and DGF was observed in 45% of recipients.

In comparison, cluster 3 was characterized by younger (median age 42 (IQR 36–50) years) black recipients who had hypertension-related kidney disease (41%) or glomerular disease (33%). Only 15% of cluster 3 recipients had diabetes, and the majority had >3 years of time on dialysis. Median BMI in cluster 3 was 42.0 (IQR 40.7–43.8) kg/m^2^. Cluster 3 recipients received young (median donor age 29 (IQR 22–38) years) standard KDPI kidneys (99%). Median CIT for cluster 3 was 14.7 (IQR 10.3–20.1) hours, and DGF was observed in 32% of recipients.

Cluster 4 was characterized by recipients with varying dialysis time (23% preemptive, 17% <1 year, 28% 1–3 years, and 31% >3 years). Cluster 4 recipients primarily had diabetes (32%) or glomerular disease (25%) as a cause of their end-stage kidney disease, and 40% were diabetic. Of the cluster 4 recipients, 49% received non-ECD deceased donor kidneys and 48% received living donor kidney transplants. Median CIT was 7.8 (IQR 1.3–16.5) hours and 15% of recipients had DGF.

Cluster 5 was characterized by kidney retransplant recipients. The majority had >3 years of dialysis (47%). Cluster 5 patients had higher PRA (median 85% (IQR 21–98%)). Of the cluster 5 patients, 69% received non-ECD deceased donor kidneys and 27% received living donor kidney transplants. Median CIT was 13.1 (IQR 4.5–20.2) hours, 30% of donor kidneys were placed on machine perfusion, and DGF was observed in 36% of recipients.

Appendix A showed the proportion of the assigned clusters based on the UNOS regions. Region 7 had the highest number of kidney transplants in morbidly obese patients. Regions 7 and 11 had the highest and lowest proportion of cluster 1, respectively. Regions 5 and 4 had the highest and lowest proportion of cluster 2, respectively. Regions 11 and 7 had the highest and lowest proportion of cluster 3, respectively. Regions 7 and 6 had the highest and lowest proportion of cluster 4, respectively. Regions 6 and 11 had the highest and lowest proportion of cluster 5, respectively.

### 3.2. Post-Transplant Outcomes of Morbid Obesity Clusters

Table 2 shows cluster-based post-transplant outcomes. One-year patient survival in clusters 1, 2, 3, 4, and 5 was 98.0%, 94.4%, 98.5%, 98.7%, and 97%, respectively. Five-year patient survival by cluster was 90.2%, 75.6%, 91.2%, 82.8%, and 86.1%. Cluster 2 had the lowest one- and five-year patient survival (Figure 4). One-year death-censored graft survival by cluster was 98.1%, 93.0%, 96.1%, 98.8%, and 93.0%. Five-year death-censored graft survival was 90.2%, 83.2%, 81.6%, 90.8%, and 77.3%. Clusters 2 and 5 had the lowest one-year death censored graft survival. At five years, clusters 5 and 3 had the lowest death-censored graft survival. There was no difference in risk of one-year allograft rejection among these five clusters (Appendix A).

## 4. Discussion

BMI is a non-specific measure of health, and reported adverse risks associated with high BMI among kidney transplant patients are variable depending on patient characteristics [7,8]. Nevertheless, given the possible increased perioperative complications and the reduced patient and allograft survival among patients with high BMI [1,2,3,4,5,6], the Scientific Registry of Transplant Recipients (SRTR) includes adjustment parameters for BMI > 25 and BMI > 30 kg/m^2^ for risk adjustment equations of transplant outcomes [28]. Despite these concerns, outcome data among morbidly obese kidney transplant patients remain limited [10,29,30,31], and kidney transplant surgeries are rarely offered to patients with a BMI of ≥40 kg/m^2^ [9,32], despite significant survival benefits of kidney transplantation versus remaining on dialysis. In this study, we utilized unsupervised ML consensus clustering and successfully identified five groups of morbidly obese kidney transplant recipients in the USA. Although the number of kidney transplant recipients with a BMI of ≥40 kg/m^2^ at the time of kidney transplant remains small, accounting for only 2.0% of all kidney transplant recipients, these five clusters of recipients have distinct characteristics and different post-transplant outcomes (Table 1).

In brief, cluster 1 recipients were predominantly white and non-sensitized, had short dialysis time or were preemptive, and were more likely to receive living donor kidney transplants. Cluster 2 recipients were older and diabetic. They were likely to have been on dialysis for >3 years and receive a standard KDPI deceased donor kidney. Cluster 3 recipients were young, black, and had kidney disease secondary to hypertension or glomerular disease, and they had >3 years of dialysis and received non-ECD, young, deceased donor kidney transplants with a KDPI < 85%. Cluster 4 recipients were likely to be diabetic with variable dialysis duration and either received non-ECD standard KDPI kidneys or living donor kidney transplants. Lastly, cluster 5 recipients were young retransplants that were sensitized

Patient survival at one year was favorable in all clusters, with cluster 2 showing the lowest survival at 94.4%. At five years post-transplant, patient survival was highest for clusters 3 and 1 and lowest for cluster 2. The reduced long-term survival observed in cluster 2 is likely a reflection of diabetes-related comorbidities, similar to what is observed in non-obese older diabetic kidney transplant recipients [33,34]. Similar to patient survival, one-year death-censored graft survival was also favorable among all five clusters. Clusters 2 and 5 had the lowest one-year death-censored graft survival. Although cluster 3 patients were the youngest among the five groups, and had best long-term survival, graft outcomes were discordant, with these recipients having the second to lowest graft survival at five years. This finding likely reflects ongoing health inequities and disparities in transplant that disproportionately impact minorities. Cluster 3 recipients were young, black, and had hypertension or glomerular-related kidney disease. The reduced graft survival in cluster 3 recipients underscores at-risk disparities such as socioeconomic post-transplant support and minorities being disadvantaged in access to preemptive and living donor transplants [35,36]. In contrast, the majority of cluster 1 and 4 recipients were white. These clusters had the greatest number of preemptive and living donor transplants. These clusters also had the lowest number of recipients with >3 years of time on dialysis. Despite approximately 40% of cluster 1 and 4 recipients having diabetes, short- and long-term patient and graft survival outcomes were more favorable, likely as a result of the shorter dialysis time and access to living donation.

Despite the use of SRTR data, there are several limitations to this study. The data shown here only represent obese patients with a BMI ≥ 40 kg/m^2^ who were transplanted. Data specific to patients with a high BMI but denied approval for transplant or approved but not transplanted remain unavailable. Kidney transplantation in recipients with a BMI ≥ 40 kg/m^2^ remains uncommon (2%), and the favorable outcomes shown in this cohort represent a highly selected group of individuals. As such, the applicability of these outcomes in other obese kidney failure patients may not be universal. Nonetheless, data specific to kidney transplant recipients with a BMI ≥ 40 kg/m^2^ remain under-reported. To our knowledge, this is the first application of a ML clustering approach specific to obese kidney transplant recipients, and application of the ML clustering approach provides individualized guidance to optimize kidney transplant for recipients with morbid obesity. The favorable outcomes shown here can hopefully provide some additional assurance to the transplant community and improve access to transplant for these patients. Future studies are needed to individualize pre- and post-transplant care to optimize the outcomes among different clusters of transplant recipients with a BMI ≥ 40 kg/m^2^.

## 5. Conclusions

In summary, kidney transplantation in those with a BMI ≥ 40 kg/m^2^ remains very uncommon. Within this small cohort, five unique phenotypic clusters of morbidly obese kidney transplant recipients were identifiable through ML clustering. Similar to nonobese kidney transplant recipients, recipients with earlier access to transplant and living donation show superior outcomes, and disparities exist for minorities. Although the identified clusters differed in their post-transplant outcomes, patient and graft survival were overall favorable in all five clusters. Kidney transplant recipients with BMI of ≥40 kg/m^2^ have variable characteristics and should receive individualized pre- and post-transplant outcomes counseling.

## Figures and Tables

**Figure 1 jcm-11-03288-f001:**
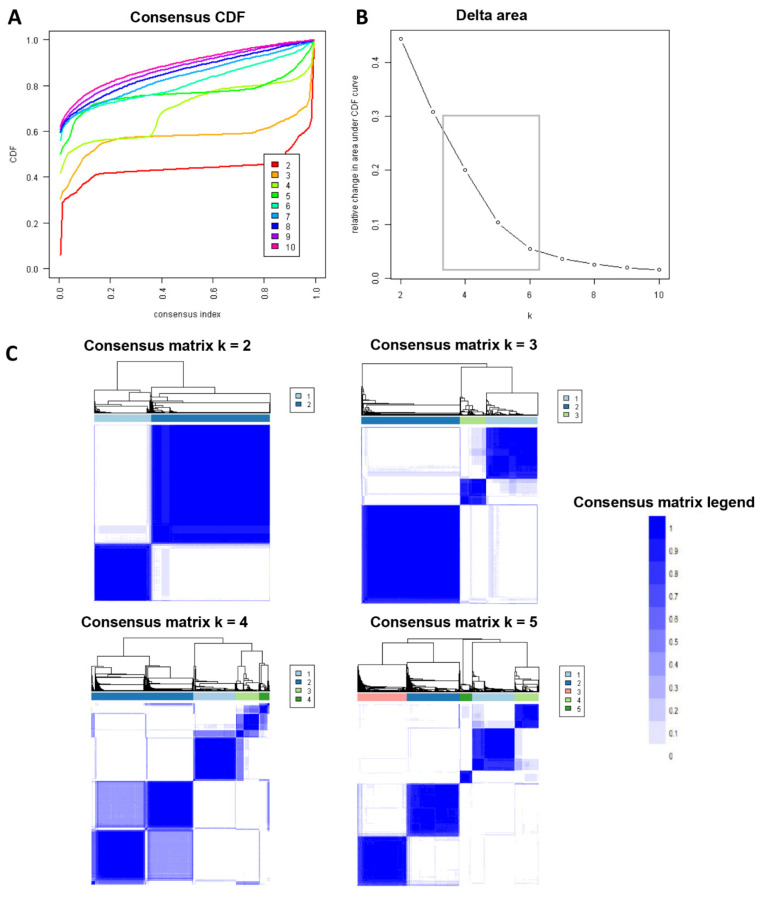
(**A**) CDF plot displaying consensus distributions for each k. (**B**) Delta area plot reflecting the relative changes in the area under the CDF curve. (**C**) Consensus matrix heat map depicting consensus values on a white to blue color scale of each cluster.

**Figure 2 jcm-11-03288-f002:**
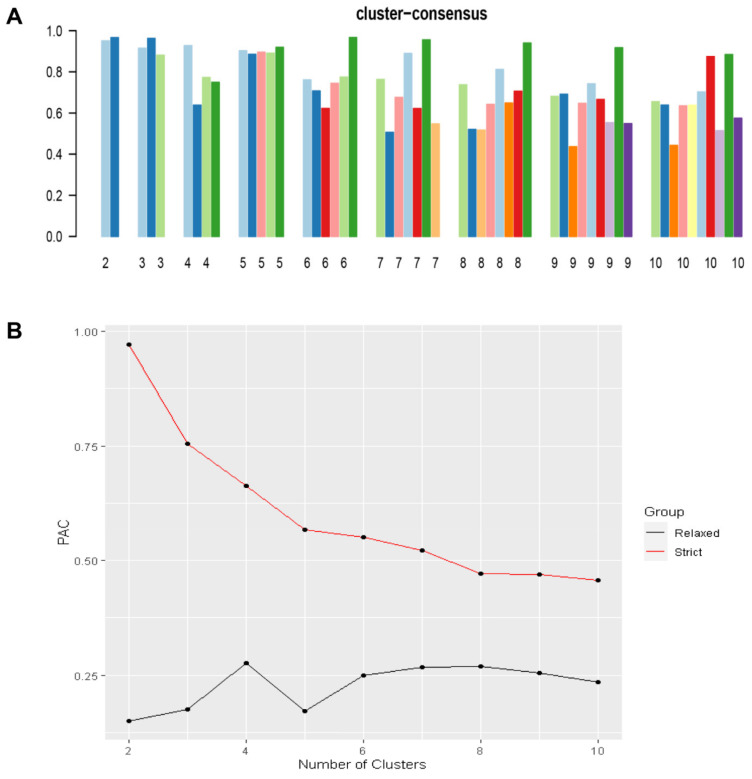
(**A**) The bar plot represents the mean consensus score for different numbers of clusters (k ranges from two to ten). (**B**) The PAC values assess ambiguously clustered pairs.

**Figure 3 jcm-11-03288-f003:**
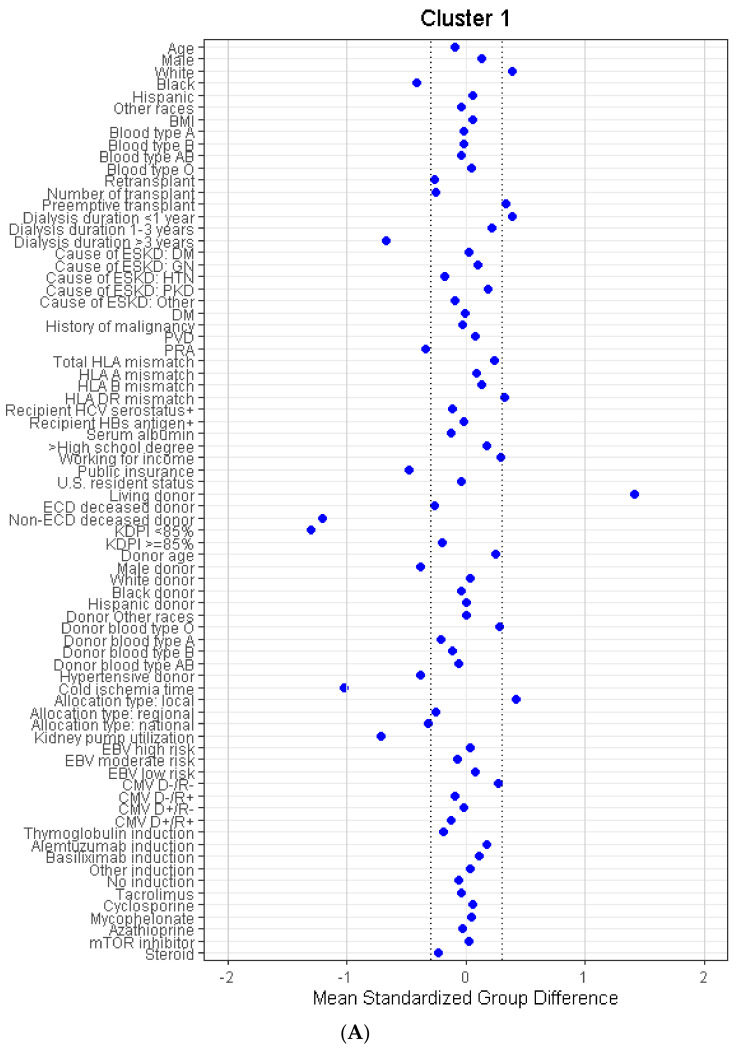
(**A**) The standardized differences in cluster 1 of morbidly obese kidney transplant recipients for each of the baseline parameters. (**B**) The standardized differences in cluster 2 of morbidly obese kidney transplant recipients for each of the baseline parameters. (**C**) The standardized differences in cluster 3 of morbidly obese kidney transplant recipients for each of the baseline parameters. (**D**) The standardized differences in cluster 4 of morbidly obese kidney transplant recipients for each of the baseline parameters. (**E**) The standardized differences in cluster 5 of morbidly obese kidney transplant recipients for each of the baseline parameters. The x axis shows the standardized differences values, and the y axis shows baseline parameters. The dashed vertical lines represent the standardized differences cutoffs of <−0.3 or >0.3. Abbreviations: BMI: body mass index, CMV: cytomegalovirus, D: donor, DGF: delayed graft function, DM: diabetes mellitus, EBV: Epstein-Barr virus, ECD: extended criteria donor, ESKD: end-stage kidney disease, GN: glomerulonephritis, HBs: hepatitis B surface, HCV: hepatitis C virus, HIV: human immunodeficiency virus, HLA: human leukocyte antigen, HTN: hypertension, KDPI: kidney donor profile index, mTOR: mammalian target of rapamycin, PKD: polycystic kidney disease, PRA: panel reactive antibody, PVD: peripheral vascular disease, R: recipient.

**Figure 4 jcm-11-03288-f004:**
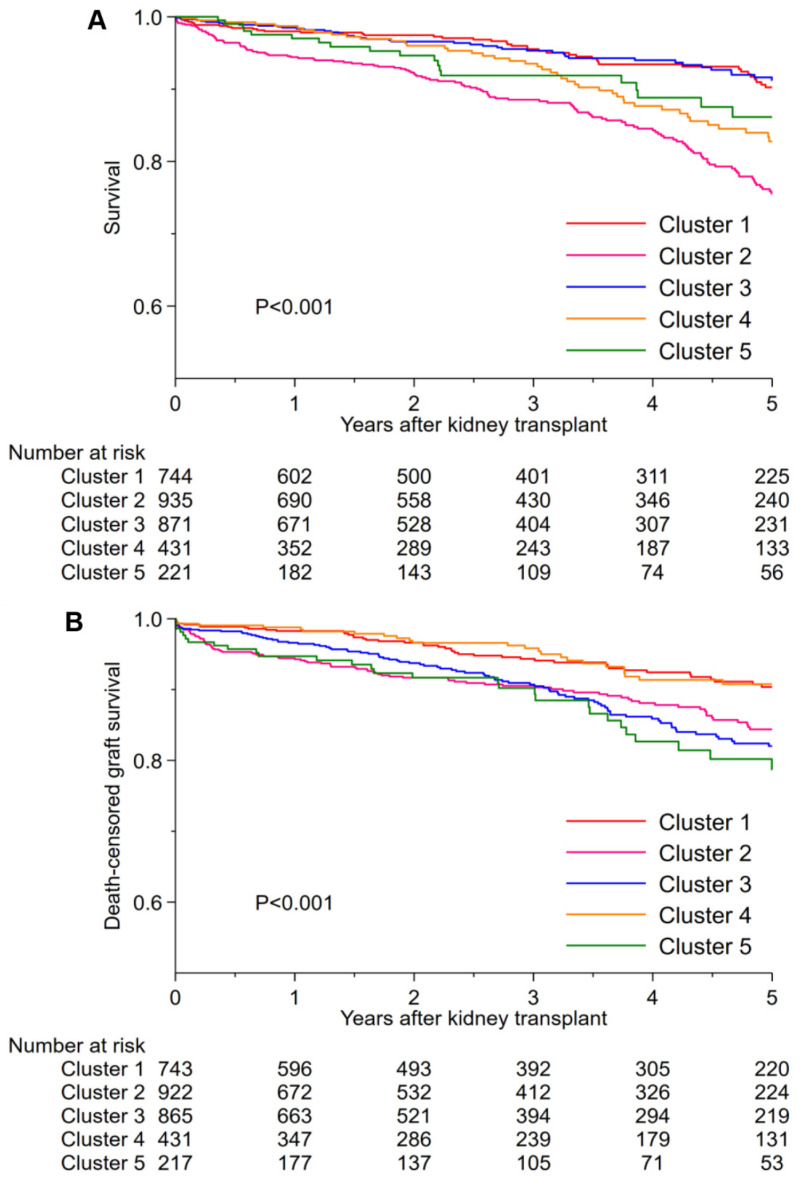
(**A**) Patient survival and (**B**) death-censored graft survival after kidney transplant among five unique clusters of morbidly obese kidney transplant recipients in the USA.

**Table 1 jcm-11-03288-t001:** Clinical characteristics according to clusters of morbidly obese kidney transplant recipients.

	All(*n* = 3204)	Cluster 1(*n* = 745)	Cluster 2(*n* = 936)	Cluster 3(*n* = 871)	Cluster 4(*n* = 431)	Cluster 5(*n* = 221)	*p*-Value
Recipient age (year), median (IQR)	50 (41–59)	49 (40–57)	58 (51–64)	42 (36–50)	51 (41–59)	46 (39–54)	<0.001
Recipient male sex	1724 (54)	452 (61)	489 (52)	462 (53)	215 (50)	106 (48)	<0.001
Recipient race							<0.001
-White	1482 (46)	489 (66)	386 (41)	218 (25)	285 (66)	104 (47)
-Black	1278 (40)	145 (19)	430 (46)	542 (62)	76 (18)	85 (38)
-Hispanic	328 (10)	90 (12)	81 (9)	73 (9)	58 (13)	26 (12)
-Other	116 (4)	21 (3)	39 (4)	38 (4)	12 (3)	6 (3)
ABO blood group							<0.001
-A	1193 (37)	270 (36)	327 (35)	307 (35)	202 (47)	87 (39)
-B	439 (14)	98 (13)	132 (14)	132 (15)	55 (13)	22 (10)
-AB	156 (5)	30 (4)	57 (6)	42 (5)	21 (5)	6 (3)
-O	1416 (44)	347 (47)	420 (45)	390 (45)	153 (35)	106 (48)
Body mass index (kg/m^2^), median (IQR)	41.7(40.6–43.4)	41.8(40.7–43.6)	41.4(40.5–42.9)	42.0(40.7–43.8)	41.5(40.6–43.1)	41.5(40.6–43.4)	<0.001
Retransplant	231 (7)	3 (0.4)	1 (0.1)	0 (0)	6 (1)	221 (100)	
Dialysis duration							<0.001
-Preemptive	447 (14)	191 (26)	64 (7)	62 (7)	100 (23)	30 (14)
-<1 year	399 (12)	189 (25)	59 (6)	49 (6)	74 (17)	28 (13)
-1–3 years	643 (20)	214 (29)	125 (13)	124 (14)	122 (28)	58 (26)
->3 years	1715 (54)	151 (20)	688 (74)	636 (73)	135 (31)	105 (47)
Cause of end-stage kidney disease							<0.001
-Diabetes mellitus	1001 (31)	243 (33)	532 (57)	79 (9)	136 (32)	11 (5)
-Hypertension	777 (24)	124 (17)	192 (20)	358 (41)	84 (19)	19 (9)
-Glomerular disease	740 (23)	204 (27)	100 (11)	286 (33)	107 (25)	43 (19)
-PKD	262 (8)	98 (13)	50 (5)	60 (7)	49 (11)	5 (2)
-Other	424 (13)	76 (10)	62 (7)	88 (10)	55 (13)	143 (65)
Comorbidity							
-Diabetes mellitus	1341 (42)	311 (42)	654 (70)	135 (15)	172 (40)	69 (31)	<0.001
-Malignancy	168 (5)	34 (5)	69 (7)	25 (3)	24 (6)	16 (7)	<0.001
-Peripheral vascular disease	280 (9)	82 (11)	117 (12)	25 (3)	40 (9)	16 (7)	<0.001
PRA (%), median (IQR)	0 (0–17)	0 (0–0)	0 (0–5)	0 (0–17)	0 (0–37)	85 (21–98)	<0.001
Positive HCV serostatus	79 (2)	5 (1)	41 (4)	20 (2)	5 (1)	8 (4)	<0.001
Positive HBs antigen	31 (1)	6 (1)	6 (1)	13 (1)	3 (1)	3 (1)	0.35
Positive HIV serostatus	19 (1)	2 (0)	7 (1)	9 (1)	1 (0)	0 (0)	0.14
Functional status							<0.001
-10–30%	12 (0)	2 (0)	2 (0)	3 (0)	3 (1)	2 (1)
-40–70%	1307 (41)	240 (32)	466 (50)	363 (42)	155 (36)	83 (38)
-80–100%	1885 (59)	503 (67)	468 (50)	505 (58)	273 (63)	136 (61)
Working income	1137 (35)	370 (50)	202 (22)	300 (34)	175 (41)	90 (41)	<0.001
Public insurance	2270 (71)	366 (49)	780 (83)	707 (81)	253 (59)	164 (74)	<0.001
US resident	3191 (99)	740 (99)	935 (100)	869 (100)	427 (99)	220 (99)	0.14
Undergraduate education or above	1817 (57)	486 (65)	486 (52)	471 (54)	240 (56)	134 (61)	<0.001
Serum albumin (g/dL), mean (SD)	3.8 ± 0.5	3.8 ± 0.5	3.8 ± 0.5	4.0 ± 0.5	3.8 ± 0.5	3.7 ± 0.5	<0.001
Kidney donor status							<0.001
-Non-ECD deceased	1950 (61)	16 (2)	716 (76)	856 (98)	210 (49)	152 (69)
-ECD deceased	251 (8)	5 (1)	215 (23)	6 (1)	15 (3)	10 (4)
-Living	1003 (31)	724 (97)	5 (1)	9 (1)	206 (48)	59 (27)
ABO incompatibility	9 (0)	8 (1)	0 (0)	0 (0)	1 (0)	0 (0)	<0.001
Donor age (year), median (IQR)	40 (29–51)	43 (33–52)	49 (39–55)	29 (22–38)	39 (29–50)	38 (26–48)	<0.001
Donor male sex	1756 (55)	267 (36)	565 (60)	591 (68)	216 (50)	117 (53)	<0.001
Donor race							0.003
-White	2248 (70)	534 (72)	676 (72)	583 (70)	318 (74)	137 (62)
-Black	509 (16)	107 (14)	135 (14)	159 (18)	54 (12)	54 (24)
-Hispanic	370 (11)	86 (11)	96 (10)	110 (13)	51 (12)	27 (12)
-Other	77 (2)	18 (2)	29 (3)	19 (2)	8 (2)	3 (1)
History of hypertension in donor	655 (20)	37 (5)	394 (42)	114 (13)	56 (13)	54 (24)	<0.001
KDPI							<0.001
-Living donor	1003 (31)	724 (97)	5 (1)	9 (1)	206 (48)	59 (27)
-KDPI < 85	2082 (65)	21 (3)	824 (88)	861 (99)	219 (51)	157 (71)
-KDPI ≥ 85	119 (4)	0 (0)	107 (11)	1 (0.1)	6 (1)	5 (2)
HLA mismatch, median (IQR)	4 (3–5)	4 (3–5)	5 (4–5)	5 (4–5)	2 (0–2)	4 (2–5)	<0.001
Cold ischemia time (hours), median (IQR)	11.7(2.7–18.8)	1.3(0.8–2.5)	17 (12–23)	14.7(10.3–20.1)	7.8(1.3–16.5)	13.1(4.5–20.2)	<0.001
Kidney on pump	1089 (34)	2 (0)	538 (57)	396 (45)	86 (20)	67 (30)	<0.001
Delay graft function	887 (28)	38 (5)	423 (45)	282 (32)	64 (15)	80 (36)	<0.001
Allocation type							<0.001
-Local	2714 (85)	743 (100)	722 (77)	757 (87)	334 (77)	158 (71)
-Regional	189 (6)	0 (0)	102 (11)	48 (5)	21 (5)	18 (8)
-National	301 (9)	2 (0)	112 (12)	66 (8)	76 (18)	45 (20)
EBV status							0.01
-Low risk	38 (1)	15 (2)	2 (0)	11 (1)	9 (2)	1 (1)
-Moderate risk	2896 (90)	659 (88)	856 (92)	786 (90)	387 (90)	208 (94)
-High risk	270 (9)	71 (10)	78 (8)	74 (9)	35 (8)	12 (5)
CMV status							<0.001
-D-/R-	624 (20)	225 (30)	140 (15)	136 (16)	99 (23)	24 (11)
-D-/R+	773 (24)	152 (20)	232 (25)	239 (27)	90 (21)	60 (27)
-D+/R+	1131 (35)	217 (29)	352 (38)	301 (35)	158 (37)	103 (47)
-D+/R-	676 (21)	151 (20)	212 (23)	195 (22)	84 (19)	34 (15)
Induction immunosuppression							
-Thymoglobulin	1745 (54)	334 (45)	528 (56)	544 (62)	197 (46)	142 (64)	<0.001
-Alemtuzumab	658 (20)	206 (28)	171 (18)	147 (17)	99 (23)	35 (16)	<0.001
-Basiliximab	525 (16)	153 (20)	148 (16)	103 (12)	103 (24)	18 (8)	<0.001
-Other	100 (3)	28 (4)	36 (4)	19 (2)	8 (2)	9 (4)	0.09
-No induction	268 (8)	51 (7)	85 (9)	81 (9)	32 (7)	19 (9)	0.36
Maintenance immunosuppression							
-Tacrolimus	2844 (89)	652 (87)	810 (86)	797 (91)	397 (92)	188 (85)	<0.001
-Cyclosporine	77 (2)	24 (3)	23 (2)	13 (1)	10 (2)	7 (3)	0.22
-Mycophenolate	2947 (92)	694 (93)	846 (90)	804 (92)	402 (93)	201 (91)	0.20
-Azathioprine	10 (0)	1 (0)	3 (0)	1 (0)	3 (1)	2 (1)	0.17
-mTOR inhibitors	37 (1)	11 (1)	9 (1)	10 (1)	4 (1)	3 (1)	0.87
-Steroid	1939 (60)	367 (49)	594 (63)	587 (67)	231 (54)	160 (72)	<0.001

**Table 2 jcm-11-03288-t002:** Post-transplant outcomes according to the clusters of morbidly obese kidney transplant recipients.

	Cluster 1	Cluster 2	Cluster 3	Cluster 4	Cluster 5
1-year survival	98.0%	94.4%	98.5%	98.7%	97.0%
HR for 1-year death	1.62 (0.62–5.01)	4.56 (2.00–13.12)	1.20 (0.45–3.77)	1 (ref)	2.33 (0.70–8.07)
5-year survival	90.2%	75.6%	91.2%	82.8%	86.1%
HR for 5-year death	0.57 (0.38–0.87)	1.58 (1.13–2.25)	0.54 (0.35–0.82)	1 (ref)	0.92 (0.52–1.54)
1-year death-censored graft survival	98.1%	93.0%	96.1%	98.8%	93.0%
HR for 1-year death-censored graft failure	1.50 (0.57–4.67)	5.89 (2.62–16.85)	3.19(1.36–9.34)	1 (ref)	5.93 (2.30–18.25)
5-year death-censored graft survival	90.2%	83.2%	81.6%	90.8%	77.3%
HR for 5-year death-censored graft failure	1.08 (0.67–1.78)	2.22 (1.46–3.51)	2.09 (1.37–3.32)	1 (ref)	2.79 (1.67–4.74)
1-year acute rejection	8.6%	7.6%	8.0%	6.0%	9.5%
OR for 1-year acute rejection	1.46 (0.91–2.35)	1.28 (0.80–2.03)	1.36 (0.85–2.17)	1 (ref)	1.64 (0.90–2.97)

## Data Availability

Data are available upon reasonable request to the corresponding author.

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
