# Peer review of "Machine Learning Consensus Clustering of Morbidly Obese Kidney Transplant Recipients in the United States"

_jcm, 2022, doi:10.3390/jcm11123288_

Round 1

Reviewer 1 Report

The purpose of this study was to use the machine learning clustering of the data of UNOS database from 2010 through 2019 to identify distinct clusters of morbidly obese kidney transplant recipients and assess clinical outcomes among them.

In material and methods section the authors should add additional information on the methods of indication on cluster stability and the way of consensus statement.

The Authors emphasized in the differences among 5 clusters.

In fact patients with BMI> 40 kg/m2 which are at the highest risk of the procedure of kidney transplantation as were as the negative outcomes related to obesity constituted only small amount of overall KT recipients. 

The very interesting results of this study is inconclusive in clinical practise because of the fact the the cohort of those small and as indicated by the authors they should receive individualized pre- and post-transplant  care to optimize of the outcome. This should be the topic of next studies and the machine learning can be in future use for indicating the risk group among morbidly obesed kidney transplant recipients. This should be emphazised in the discussion section.

Minor language errors should be improved.

Author Response

Response to Reviewer#1

The purpose of this study was to use the machine learning clustering of the data of UNOS database from 2010 through 2019 to identify distinct clusters of morbidly obese kidney transplant recipients and assess clinical outcomes among them.

Response: Thank you for reviewing our manuscripts and your critical evaluation.

Comment #1

In material and methods section the authors should add additional information on the methods of indication on cluster stability and the way of consensus statement.

Response: We agree with the reviewer. We have added additional information on the methods of indication on cluster stability and the way of consensus statement as the reviewer’s suggestions. We also added more details in online supplementary.

“The within-cluster consensus score, ranging between 0 and 1, was defined as the average consensus value for all pairs of individuals belonging to the same cluster [26]. A value closer to one indicates better cluster stability. PAC, ranging between 0 and 1, was calculated as the proportion of all sample pairs with consensus values falling within the predetermined boundaries [27]. A value closer to zero indicates better cluster stability [27].”

Comment #2

In fact patients with BMI> 40 kg/m2 which are at the highest risk of the procedure of kidney transplantation as were as the negative outcomes related to obesity constituted only small amount of overall KT recipients.

The very interesting results of this study is inconclusive in clinical practise because of the fact the the cohort of those small and as indicated by the authors they should receive individualized pre- and post-transplant  care to optimize of the outcome. This should be the topic of next studies and the machine learning can be in future use for indicating the risk group among morbidly obesed kidney transplant recipients. This should be emphazised in the discussion section.

Response: We agree with the reviewer. We have additionally included this important point in the discussion as suggested.

“Future studies are needed to individualize pre- and post-transplant care to optimize the outcomes among different clusters of transplant recipients with a BMI >40 kg/m2.”

Comment #3

Minor language errors should be improved.

Response: We appreciate the reviewer’s thorough reviews. We have reviewed and corrected the minor errors throughout manuscript.

Reviewer 2 Report

The authors have presented a well written and original study on a subject that is underreported. The data is well presented and its interpretation logical. Very minor editing of text/spelling required.

Author Response

Response to Reviewer#2

Comment

he authors have presented a well written and original study on a subject that is underreported. The data is well presented and its interpretation logical. Very minor editing of text/spelling required.

Response: Thank you for reviewing our manuscripts and your critical evaluation. The error has been corrected.

Thank you for your time and consideration.  We greatly appreciated the reviewer's and editor's time and comments to improve our manuscript. The manuscript has been improved considerably by the suggested revisions.
